# Substitution of surgical care within benign gynaecology during COVID-19: waste of a good crisis? – a quantitative longitudinal study in the Netherlands

Eva L M Velthuijs ![ORCID] ,[1] Ismail Ismail,[2] Xander Koolman,[2] Robert A de Leeuw,[1] Wouter J K Hehenkamp[1]

¹Department of Gynaecology and Obstetrics, Amsterdam UMC Locatie AMC, Amsterdam, The Netherlands
²Department of Health Economics, Vrije Universiteit Amsterdam, Amsterdam, The Netherlands

**Correspondence to**
Eva L M Velthuijs;
e.velthuijs@amsterdamumc.nl

## ABSTRACT

**Objective** To examine the impact of the COVID-19 pandemic on the substitution of surgical procedures in benign gynaecology in the Netherlands.

**Design** Quantitative longitudinal study evaluating the effects of the COVID-19 pandemic.

**Setting** Nationwide healthcare delivery was analysed across six benign gynaecological pathways from 2016 to 2022 using Vektis and Dutch Hospital Data (DHD), accessed via Statistics Netherlands (Centraal Bureau voor de Statistiek).

**Participants** The study focused on six benign gynaecological pathways classified using Dutch Diagnosis Treatment Combinations (DTCs): heavy menstrual blood loss (G11), uterine fibroids (G15), endometriosis (G17), prolapse (G25), infertility treatment (F11) and first trimester pregnancy complications (Z12). All patients receiving care within these pathways between 1 January 2016 and 31 December 2022 were included. Exclusions applied to all patients under 18 years old and, only within the menstrual disorder pathway, patients over 51 years old to exclude most postmenopausal blood loss cases where no alternative treatment applies.

**Interventions** Cohorts from the initial pandemic year (2020) were compared with four prepandemic cohorts (2016–2019) and late-pandemic (2021) and postpandemic (2022) cohorts.

**Primary and secondary outcome measures** The primary outcome was the trend in the total number of patients in surgical and non-surgical procedure groups across cohort periods. Secondary outcomes included trends within individual pathways.

**Results** The analysis identified a significant reduction in benign gynaecological care during 2020, with an 18.3% (p<0.001) decrease in surgical procedures and a 6% (p=0.02) decrease in non-surgical procedures across all pathways. During the COVID-19 pandemic, surgical care showed the largest significant declines in infertility treatment (−19.3%, p<0.001) and prolapse and heavy menstrual bleeding (both −17.5%, p<0.001), while non-surgical care decreased less markedly, with the largest significant drop in prolapse care (−10.4%, p=0.001). Non-surgical procedures generally returned to pre-pandemic levels, whereas surgical care remained reduced. These differences were conducted using regression models that adjusted for time trends and the COVID-19-related impact.

## STRENGTHS AND LIMITATIONS OF THIS STUDY

⇒ The large dataset spanning allows a detailed longitudinal insight into gynaecological care trends before, during and after the COVID-19 pandemic.

⇒ Examining multiple gynaecological care pathways offers a detailed understanding of how different clinical contexts were affected by the pandemic.

⇒ The observational nature of the study limits our ability to infer causality between the pandemic and the changes in care utilisation.

**Conclusions** The COVID-19 pandemic significantly disrupted both surgical and non-surgical procedures within benign gynaecological pathways. Reduced care uptake during the pandemic waves was not recovered but instead forgone. The reduction in surgical procedures did not correspond with increased use of non-surgical alternatives. Future research should prioritise evaluating the long-term impacts of this disruption on patients and society.

## INTRODUCTION

Healthcare systems are under increasing strain due to rising expenditures and limited resources.[1][2] This results in a need for more sustainable and efficient care models, which is increasing over time. Substitution of high-intensity interventions, such as surgical procedures, with less resource-intensive alternatives may help if they result in adequate outcomes.[3] However, such substitution is often hindered by concerns about clinical efficacy and organisational resistance.[4]

The COVID-19 pandemic, with its widespread disruption of elective surgical procedures and resource allocation towards urgent and critical care, created a natural experiment to examine whether substitution increased in times of acute and prolonged scarcity.[5–8] The initial COVID-19 period showed a reduction of 37% of all healthcare services across 20 countries.[9] Non-COVID-19

hospital admissions were reduced by 28%. In the Netherlands, mainly non-cancer surgical treatments were cancelled during COVID-19 peaks.[10] National policies restricted non-urgent hospital care, especially during the first wave (March–May 2020), the second wave (October 2020–February 2021) and subsequent surges in 2021. Formal triage measures for surgical care, led by the Dutch Federation of Medical Specialists, were implemented to determine which procedures should be postponed.[11] Restrictions were lifted gradually during 2022 as hospital pressures subsided. The combination of a sharp reduction in surgical capacity and a lacking perspective to recover delayed care created a window of opportunity for less capacity-consuming, less invasive (non-) surgical alternatives. While most research to date studies the impact of the reduction of surgical care, none describe the effect of the pandemic on substitution.[9 12–15]

Within gynaecology, elective surgical procedures were significantly curtailed during the COVID-19 pandemic due to the limited surgical capacity and resource allocation towards urgent and critical care.[16–19] In benign gynaecology, surgical interventions, such as hysterectomies, are used to treat conditions such as bleeding disorders and uterine fibroids.[20] While these surgeries effectively halt disease progression and improve quality of life, less invasive treatments like hormonal therapies and outpatient procedures are underused as an alternative, leading to unnecessary invasive and costly care.[21] While shifts towards less invasive gynaecological care have long been called for, actual substitution rates remained low before the pandemic.[22 23]

Given the growing scarcity of healthcare resources globally,[24] this study aims to describe the pandemic's impact on surgical procedures and non-surgical care in benign gynaecology in the Netherlands. We hypothesise that acute surgical capacity constraints of labour-intensive surgical procedures were substituted by more efficient non-surgical alternatives across the six chosen pathways during the initial COVID-19 period due to resource shortages and the need for alternative treatment strategies.[12 25]

Our goal is to uncover whether, and to what extent, substitution occurred and was sustained during and after this capacity squeeze of the operation theatre for an extended period. In addition, we will describe patterns of healthcare deferral and the extent to which it was forgone for each of the pathways.

## METHODS
### Study design
This quantitative longitudinal study describes the use of benign gynaecological care pathways in the Netherlands before, during and after the COVID-19 pandemic. A care pathway encompasses a disease and all associated care products and activities, providing a comprehensive framework for managing the patient's entire course of care. We compared the initial pandemic year (2020) to four pre-pandemic years (2016–2019), as well as a late pandemic (2021) and a post-pandemic (2022) year.

The study focuses on six benign gynaecological pathways, classified using Dutch Diagnosis Treatment Combinations (DTCs): heavy menstrual bleeding (G11), uterine fibroids (G15), endometriosis (G17), prolapse (G25), infertility treatment (F11) and first trimester pregnancy complications (Z12). These conditions were selected because they include both elective surgical and non-surgical treatment options with comparable outcomes, allowing for potential substitution during periods of surgical capacity restriction.[26–32]

### Data collection
Healthcare delivery across the six benign gynaecological pathways from 2016 to 2022 was analysed using Vektis and Dutch Hospital Data (DHD), accessed through Statistics Netherlands (Centraal Bureau voor de Statistiek, CBS). Vektis is the Dutch healthcare information centre that collects and analyses health insurance claims to support transparency and improve healthcare quality and efficiency. DHD manages and provides hospital data derived from electronic health records, including patient admissions, diagnoses, procedures and other care activities. These datasets together provide near-complete coverage of hospital-based gynaecological care in the Netherlands. DHD captures virtually all hospital claims data, while Vektis includes claims from all Dutch health insurers. CBS, the national statistical office, provides comprehensive data on a wide range of topics, including healthcare. CBS data integrate with Vektis and DHD, offering broader insights into healthcare performance and population health.

Pathway identification was based on DTCs. Both surgical and non-surgical procedures were identified through healthcare products and activity numbers linked to individual patient records. The categorisation of procedures as surgical or non-surgical was determined by these healthcare products and activity numbers. Surgical procedures are defined as procedures that require an anaesthesiologist present, typically in the operation room. Non-surgical procedures include less invasive alternatives that do not. Procedures were only included if both product and activity codes consistently belonged to the same category. Records with inconsistent classification were excluded. The introduction date of a new DTC marked the beginning of surgical or non-surgical care. Table 1 provides an overview of the classification and distribution of procedures within the selected DTCs.

### Population
Patients who received care within one of the specified gynaecological care pathways from 1 January 2016 to 31 December 2022 were included. Care contains the overall healthcare products, whether they are part of a DTC or labelled as 'other' and are given a start and end date. Exclusions were made for patients under 18 years old and, specifically within the menstrual disorder pathway,

**Table 1** Distribution of procedures in surgical or non-surgical care

| Care pathway | Surgical | Non-surgical |
|---|---|---|
| Heavy menstrual bleeding | Hysterectomy (open/laparoscopic/vaginal/robot) | Endometrial ablation<br>GNRH analogues<br>Hormonal intrauterine device<br>Oral anticonception<br>NSAIDs<br>Tranexamic acid<br>Wait-and-see |
| Uterine fibroids | Hysterectomy (open/laparoscopic/vaginal/robot)<br>Myomectomy<br>Trans cervical removal of myoma | Uterine artery embolisation<br>GNRH analogues<br>Hormonal intrauterine device<br>Oral contraception<br>NSAIDs<br>Tranexamic acid<br>Wait-and-see |
| Endometriosis | Laparoscopic or laparotomic debulking of endometriosis | Hormonal intrauterine device<br>Oral contraception<br>Wait-and-see |
| Prolapse | Prolapse surgery | Pelvic physiotherapy<br>Pessaries<br>Wait-and-see |
| First trimester pregnancy complications | Curettage<br>Laparoscopic or laparotomic ectopic pregnancy surgery | Misoprostol<br>Methotrexate treatment<br>Wait-and-see |
| Infertility treatment | Diagnostic/therapeutic laparoscopic surgery | Intrauterine insemination<br>In Vitro Fertilisation<br>Wait-and-see |

GNRH, gonadotropin hormone-releasing hormone; NSAID, Non-steroidal anti-inflammatory drug.

for those over 51 years old to exclude most postmenopausal bleeding cases where no alternative treatment applies.

### Dataset processing

To create our dataset, several DHD and Vektis sources were combined within the CBS microenvironment. Combining, processing and cleaning of the datasets is done using RStudio. Due to mandatory national reporting, data completeness was high, and no imputation was needed. Datasets were linked via anonymous patient identifiers shared across sources. The final dataset consisted of patient-level rows with columns for DTC, care product, care activity, care start and end date, and age. Inclusion and exclusion criteria were applied accordingly. Our dataset includes all Dutch patients who received care within the six selected pathways between 2016 and 2022.

### Outcome measures

Healthcare use during the pandemic year (2020) was compared with levels and trends from 2016 to 2019, 2021 and 2022. To account for seasonal variation and for graphical presentation only, a 3-month moving average was used to monthly absolute patient counts. Each point in the moving average represents the average of one preceding, the current and a subsequent month, to reduce noise. A predicted trendline for the years 2020–2022 was estimated to serve as a counterfactual, representing a scenario without COVID-19. This prediction was constructed by calculating the average annual growth rate in patient volumes between 2016 and 2019, and applying this rate to the baseline monthly volumes of 2019. This approach is an extrapolation model based solely on monthly data in patient volumes. The primary outcome was the trend in total patient numbers in surgical and non-surgical procedure groups across the cohort periods. Secondary outcomes included trends within individual pathways.

### Statistical analysis

Statistical analyses were conducted to test the significance of observed differences in descriptive data across years. To assess raw associations, we used linear regression for continuous outcomes. To express regression outcomes in percentages, all continuous outcomes were log-transformed prior to analysis. When the distribution of the error term was not normal, we used non-parametric bootstrap to compute p values. Where relevant, year-to-year differences in proportions were tested using $\chi^2$ tests. Statistical significance was set at p value <0.05. In the linear regression models, 2016 was used as the reference year. The year 2020 was included as a dummy variable to account for the impact of the COVID-19 pandemic. Additionally, a dummy variable for surgical care was included

to differentiate between operative and non-operative care within the model. This enabled analysis of both pandemic and procedure type effects. Missing data were minimal and no imputation was performed. Data were analysed using RStudio (V.4.3.0).

## Patient and public involvement

There was no patient or public involvement in the design, conduct, reporting or dissemination of this research. The study was conducted solely by the research team, and there were no consultations with patients or members of the public during its planning or execution.

## RESULTS

A total of 2 281 613 patient records across six care pathways were included for the years 2016–2022 in the Netherlands. Table 2 shows the number of new records and the proportion of surgical care for each pathway over this period.

Overall, patient records increased from 320 157 in 2016 to 335 006 in 2022, representing a 4.6% rise. The highest number of records was in 2022 (335 006), while the lowest was in 2020 (309 631). The prolapse pathway had the largest volume, while endometriosis had the smallest. In 2020, the proportion of surgical care records declined across all pathways compared with 2019. The greatest decrease was in the prolapse pathway (−2.7%, p<0.001), while the smallest was in the infertility treatment pathway (−0.4%, p<0.001).

Figure 1 illustrates total patient records for surgical and non-surgical care from 2016 to 2022, with a predicted trendline from 2020 to 2022.

The predicted trendlines reveal a decrease in actual patient numbers compared with predicted patient numbers for both surgical and non-surgical procedures in the year 2020 compared with pre-COVID trends. The drop in 2020 amounts to a reduction of 6% (p=0.02) for non-surgical care group patient numbers and a reduction of 18.3% for surgical care (p<0.001 and compared with non-surgical care p=0.001). In the non-surgical care group, patient numbers were restored to predicted levels by July 2020. In the surgical care group, numbers remained below predicted trends throughout 2020 and 2021 and returned to the predicted trendline in 2022. A seasonal fluctuation is evident in the surgical care group over the years.

Figure 2 presents graphs of the six care pathways separately. The graphs show the number of patient records for both surgical and nonsurgical care from 2016 to 2022, with a predicted trendline from 2020 to 2022.

A similar trend pattern was observed across the pathways for heavy menstrual bleeding, endometriosis, uterine fibroids and prolapse. In these pathways, both surgical and non-surgical procedures showed percentage decrease during the COVID-19 period in 2020. Over time, surgical care did not return to pre-COVID trends, whereas non-surgical care did. In the pathway to heavy menstrual bleeding, the surgical procedures declined by 17.5% (p<0.001 and compared with non-surgical care p=0.01) while non-surgical procedures decreased by 5.5%, though not statistically significant (p=0.09). In the uterine fibroids pathway, surgical care was reduced by 12.6% (p<0.001 and compared with non-surgical care p=0.2) and nonsurgical care was reduced by 7.5% (p=0.01). Both types of care in the endometriosis pathway dropped in percentages: surgical care by 16.9% (p=0.08 and compared with non-surgical care p=0.1) and non-surgical care by 5.4% (p=0.03). The largest decrease is seen in the pathway prolapse, with a decline of 10.4% (p=0.001) in non-surgical procedures and 17.5% (p<0.001 and compared with non-surgical care p=0.6) in surgical procedures.

The pathways for first trimester pregnancy complications and infertility treatment exhibited a pre-COVID pattern of declining surgical procedures and stable non-surgical procedures. During the initial COVID-19 period, both types of care declined, but they returned to pre-COVID trends by around June/July 2020. The first trimester pregnancy complications pathway shows a non-significant difference of 0.6% (p=0.8) of non-surgical procedures over 2020 compared with predicted trends, while the surgical procedures declined by 17.7% (p<0.001 and compared with non-surgical care p=0.2). Infertility treatment shows a similar pattern with a non-significant 2.2% decrease in non-surgical procedures (p=0.6) and a statistically significant 19.3% reduction in surgical procedures (p<0.001 and compared with non-surgical care p=0.01).

The percentages and p values reported above are derived from regression models that adjust for time trends, intervention type (surgical vs non-surgical) and the specific impact of the year 2020 (COVID-19). Full model coefficients and interaction terms are presented in table 3.

## DISCUSSION

In this quantitative cohort study, we evaluated the potential substitution of surgical procedures with non-surgical procedures across six gynaecological care pathways during the initial phase of the COVID-19 pandemic in the Netherlands. Our analysis revealed an 18.3% reduction in surgical and 6% in non-surgical care across all pathways in 2020. While non-surgical procedures resumed pre-pandemic trends by mid-2020, surgical procedures remained below baseline. This implies that reductions in healthcare in 2020 were not compensated during the duration of the study. This trend was most pronounced in the prolapse pathway and was also observed in the pathways for menstrual disorders, uterine fibroids and endometriosis.

In contrast, the care pathways for urgent first trimester pregnancy complications and infertility treatment showed pre-pandemic trends of declining surgical procedures and stable non-surgical procedures. For first trimester

**Table 2** Baseline characteristics care pathways per year

| Care pathway | 2016 | 2017 | 2018 | 2019 | 2020 | 2021 | 2022 |
|---|---|---|---|---|---|---|---|
| Menstrual disorders (n) | 89199 | 89942 | 94360 | 95111 | 88016 | 94006 | 92762 |
| Surgical care (%) Δ %* (p value) | 14.2 | 14.3+0.1 (0.55) | 14.0−0.3 (0.07) | 13.2−0.8 (<0.001) | 11.4−1.8 (<0.001) | 11.6+0.2 (0.18) | 12.0+0.4 (0.01) |
| Uterine fibroids (n) | 22350 | 22924 | 23993 | 23581 | 21776 | 23148 | 24078 |
| Surgical care (%) Δ %* (p value) | 17.9 | 17.3−0.6 (0.1) | 17.1−0.2 (0.58) | 17.2+0.1 (0.78) | 16.2−1.0 (0.005) | 16.6+0.4 (0.26) | 16.5−0.1 (0.78) |
| Endometriosis (n) | 16177 | 17291 | 19456 | 21692 | 22592 | 26532 | 29494 |
| Surgical care (%) Δ %* (p value) | 8.7 | 8.5−0.2 (0.54) | 8.7+0.2 (0.51) | 8.8+0.1 (0.74) | 7.0−1.8 (<0.001) | 6.8−0.2 (0.4) | 7.4+0.6 (0.01) |
| Prolapse (n) | 92152 | 92536 | 95888 | 97799 | 84956 | 92299 | 100561 |
| Surgical care (%) Δ %* (p value) | 12.2 | 12.1−0.1 (0.51) | 12.2+0.1 (0.51) | 11.9−0.3 (0.04) | 9.2−2.7 (<0.001) | 9.7+0.5 (<0.001) | 11.5+1.8 (<0.001) |
| First trimester pregnancy complications (n) | 39395 | 36248 | 37129 | 35809 | 34908 | 35323 | 33749 |
| Surgical care (%) Δ %* (p value) | 30.0 | 26.7−3.3 (<0.001) | 24.5−2.2 (<0.001) | 24.0−0.5 (0.12) | 21.8−2.2 (<0.001) | 20.4−1.4 (<0.001) | 20.2−0.2 (0.52) |
| Infertility treatment (n) | 60884 | 60562 | 60806 | 60583 | 57383 | 59801 | 54362 |
| Surgical care (%) Δ %* (p value) | 2.5 | 2.4−0.1 (0.26) | 2.0−0.4 (<0.001) | 1.9−0.1 (0.22) | 1.5−0.4 (<0.001) | 1.5−0 (1.0) | 1.5−0 (1.0) |
| Total (N) | 320157 | 319503 | 331632 | 334575 | 309631 | 331109 | 335006 |
| Surgical care (%) Δ %* (p value) | 13.3 | 12.7−0.6 (<0.001) | 12.4−0.3 (<0.001) | 11.9−0.5 (<0.001) | 10.1−1.8 (<0.001) | 10.1−0 (1.0) | 10.9+0.8 (<0.001) |

: Covid-19 period.
*Percentual difference from previous year.

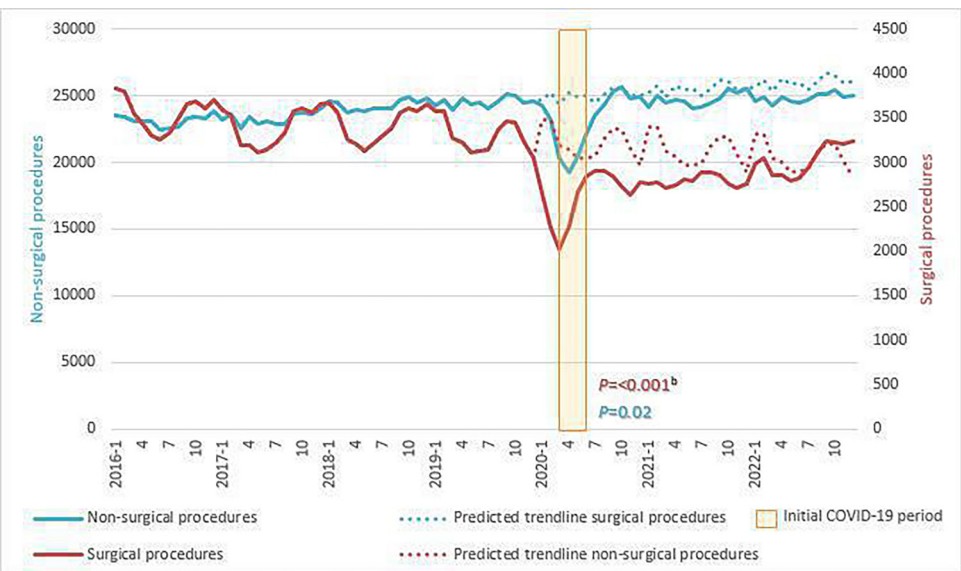

**Figure 1** Total patient count surgical and nonsurgical care timeline 2016–2022 for the six care pathways combined. [a]Trendlines are presented using 3 months moving averages. [b]Regression models were used to estimate deviations from expected trends in the year 2020; p values for this period's effect are reported within the graph in corresponding colour, full model can be found in table 3.

pregnancy complications, reductions were minimal, likely due to the urgent character of care, mirroring patterns for acute care during the pandemic.[11 33 34] Infertility care decreased early in the pandemic, with both procedural types returning to pre-pandemic patterns by mid-2020.

Our findings align with the literature documenting reductions in elective surgical care. Previous studies attributed these to the redeployment of healthcare resources towards COVID-19 care and the imposition of restrictions on elective procedures.[35] For example, a study in Japan reported an 8.5% decline in benign gynaecological surgeries in 2020 compared with 2019,[18] while New South Wales experienced a 37% reduction in general elective surgeries.[33]

Contrary to our initial hypothesis, we found no increase in substitution by non-surgical alternatives. Instead, both surgical and non-surgical procedures declined across multiple care pathways. This aligns with earlier reports of reduced non-COVID medical and surgical hospitalisations by 22% and 33%, respectively,[36] suggesting a more generalised reduction in healthcare capacity during the pandemic. This overall decrease may have resulted from a combination of reduced patient demand and reduced healthcare capacity. Data from the Dutch Healthcare Authority and ZorgDomein showed that the number of referrals during the first COVID-19 wave was 12% lower than in 2019, with a marked decline in gynaecology referrals between March and May 2020, before returning to normal by June.[37] To our surprise, no increase in referrals was observed to recover delayed care. This decline in referrals contributed to the observed reduction in both surgical and non-surgical procedures within the gynaecological care pathways,[19] but does not explain the sustained effect of continuously lower surgical procedure rates.

Patient-related factors likely influenced the reduction in referrals during the pandemic. Older patients, at higher risk for severe COVID-19 outcomes,[38] may have been more hesitant to seek care, which could explain the pronounced decline in the prolapse pathway, predominantly affecting older individuals. Symptom severity and urgency also played a role; public health campaigns discouraged in-person visits,[39–41] likely deterring patients from seeking care for non-urgent conditions such as menstrual disorders, fibroids, endometriosis and prolapse. Although stay-at-home measures may have increased awareness of symptoms, they may also have allowed patients to tolerate these symptoms more easily in a home-based setting, reducing help-seeking behaviour.

The fact that care volumes did not return to pre-pandemic levels in 2021 or 2022 suggests that much of the missed care was not simply 'caught up'. The persistent declines across all six care pathways studied may reflect structural changes in care delivery. Patients may have deferred or abandoned treatment for various reasons, such as symptom resolution, fluctuating symptoms or changing perceptions of the risks and benefits of surgery in the aftermath of the pandemic. In the case of prolapse, mortality in older patients may also have played a role. For abnormal bleeding, the natural onset of menopause may have led to symptom reduction. In addition, organisational constraints such as ongoing capacity limitations, staff shortages and prioritisation of other care domains may have contributed to sustained lower care volumes. These findings suggest that the pandemic not only caused temporary disruption in 2020 but may have accelerated or reinforced shifts in gynaecological care patterns that require long-term follow-up. Understanding whether these changes reflect appropriate care reduction, unmet

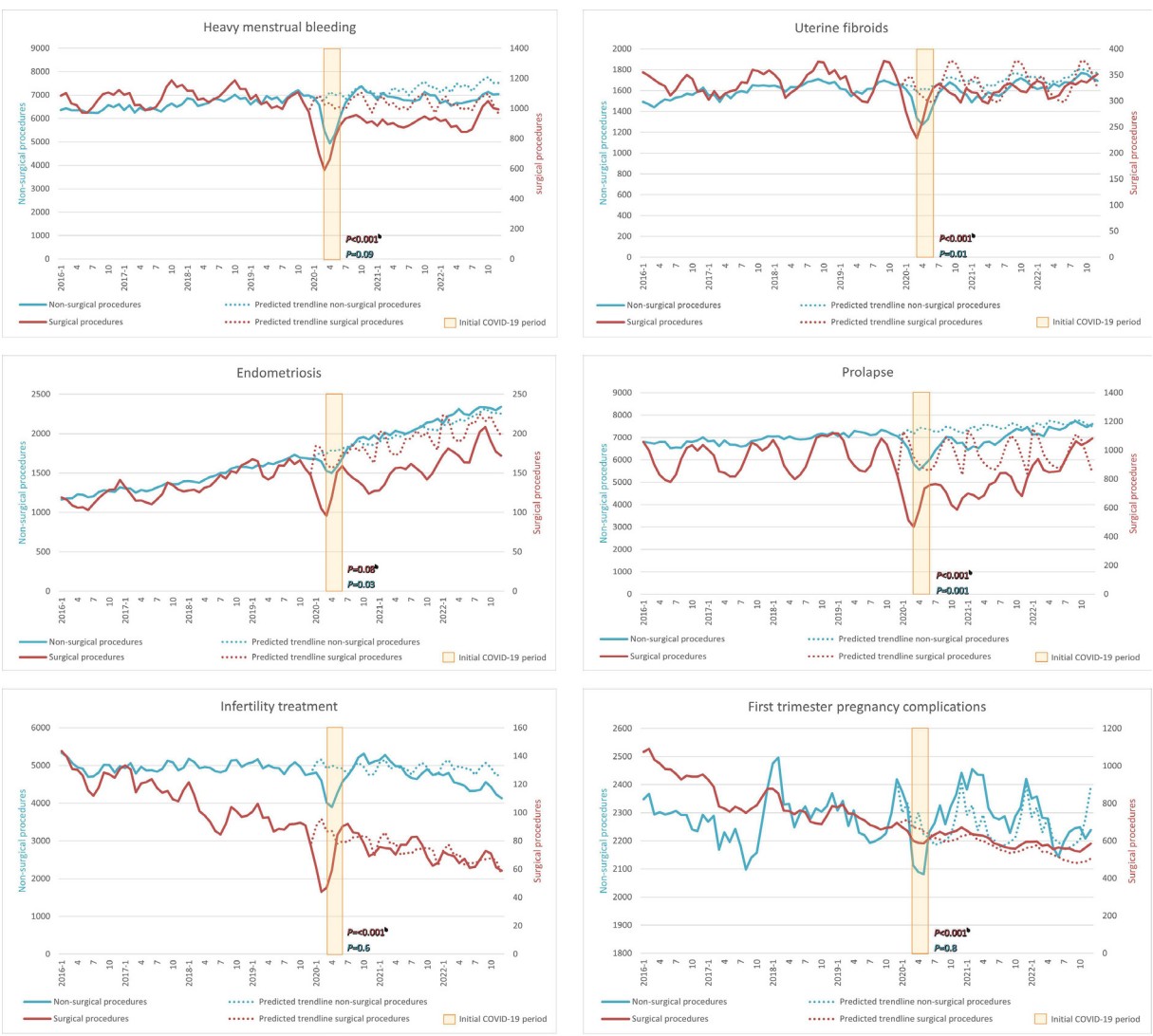

**Figure 2** Six care pathways: surgical and non-surgical care timeline 2016–2022. [a]Trendlines are presented using 3 months moving averages. [b]Regression models were used to estimate deviations from expected trends in the year 2020; p values for this period's effect are reported within the graphs in corresponding colour, full models can be found in table 3.

need or both is essential for future planning in benign gynaecology.

Professional preferences and perceptions may have contributed to treatment deferral. Gynaecologists may perceive non-surgical options as less effective than surgical procedures. For instance, uterine artery embolisation may be viewed as less reliable than hysterectomy, prompting delays until surgery was possible.[42] Previous research indicates that not all gynaecologists discuss uterine embolisation as a treatment for fibroids.[43] When surgical procedures are considered the gold standard, particularly for conditions like prolapse, fibroids or advanced endometriosis, gynaecologists may recommend delaying treatment rather than pursuing less effective non-surgical alternatives.

This inclination to defer surgery may be inappropriate in moderate to severe cases like prolapse or endometriosis, where symptom progression is likely. According to the Dutch Association of Medical Specialists (2023), patient preferences also influence treatment decisions.[44]

Some patients prefer to wait for definitive surgery after failed conservative treatments or view non-surgical options as temporary, reinforcing a shared decision to prioritise surgery even in times of resource scarcity.

At the system level, pandemic-related staff redeployment, including anaesthesiologists to COVID-19 units, reduced operating room capacity for elective procedures.[5] Gynaecological procedures like those for menstrual disorders, fibroids and prolapse were deferred, while urgent surgeries, such as those for first-trimester pregnancy complications, continued due to their non-elective nature. COVID-19-related staff absences also delayed elective surgeries and outpatient intakes.[45] Combined with patient reluctance and perceptions of non-surgical alternatives, these factors likely drove the significant procedural volume reductions observed.

### Strengths and limitations

A major strength is our large dataset spanning all benign gynaecological elective care in Dutch hospitals and clinics

**Table 3** Regression model output for yearly trends and COVID-related changes in surgical and non-surgical care across the six gynaecological care pathways

| Model: | Total‡ | Heavy menstrual bleeding‡ | Uterine fibroids‡ | Endometriosis§ | Prolapse§ | Infertility treatment‡ | First trimester pregnancy complications§ |
|---|---|---|---|---|---|---|---|
| Variables: | | | | | | | |
| Intercept | 10.053** (0.016) | 8.772** (0.020) | 7.353** (0.018) | 7.093** | 8.819** | 8.522** (0.026) | 7.729** |
| Year | 0.013** (0.005) | 0.012* (0.006) | 0.011 (0.005) | 0.106** | 0.012** | –0.013 (0.007) | 0.001 |
| OR_prestation | –1.878** (0.023) | –1.767** (0.028) | –1.539** (0.026) | –1.068** | –2.789** | –3.677** (0.037) | –1.707** |
| Int_Year_OR | –0.046** (0.007) | –0.042** (0.008) | –0.015* (0.007) | –0.139** | –0.045* | –0.096** (0.011) | –0.031 |
| Dummy_2020 | –0.060* (0.026) | –0.055 (0.032) | –0.075* (0.029) | –0.054 | –0.104** | –0.022 (0.043) | –0.006 |
| Int_OR_2020 | –0.123** (0.037) | –0.119** (0.045) | –0.051 (0.041) | –0.115 | –0.071 | –0.171** (0.060) | –0.170 |

Notes: *Intercept:* baseline level in 2016; *Year:* time trend; *OR_prestation:* effect of OR versus non-surgical care (1 = surgical); *Int_Year_OR:* interaction year × OR_prestation (surgical care); *Dummy_2020:* effect of year 2020 (COVID-19); *Int_OR_2020:* interaction OR_prestation × 2020.
SEs in parentheses.
Significance codes: **= <0.01, *= <0.05.
‡Linear regression model normally distributed.
§Linear regression model with non-parametric bootstraps R=10 000.
OR, operating room (surgical care).

over 6 years. This allows a detailed longitudinal insight into gynaecological care trends before, during and after the COVID-19 pandemic. Moreover, the DHD provider follows standardised data collection protocols that enhance the comparability of our findings. In addition, examining multiple gynaecological care pathways offers a detailed understanding of how different clinical contexts were affected by the pandemic.

Nevertheless, there are important limitations. The observational nature of the study limits our ability to infer causality between the pandemic and the changes in care utilisation. Additionally, we did not assess primary healthcare use, which might have substituted a share of the secondary care and patient outcomes, which restricts our ability to evaluate the effectiveness of the reduced or substituted non-surgical procedures. This latter limitation is particularly relevant when considering the long-term consequences of deferred surgical interventions.

### Implications for practice
Our findings have significant implications for clinical practice and healthcare policy. The observed reduction in both surgical and non-surgical procedures during the initial phase of the COVID-19 pandemic, with no evidence of subsequent compensation, suggests a fundamental shift in healthcare delivery: care was not deferred or substituted, but it appears to have vanished. This phenomenon of forgone care necessitates a deeper understanding of the underlying causes and the potential long-term consequences for patients and healthcare systems.

First, it is critical to investigate what happened to the patients who did not receive care during this period. Did they seek alternative forms of care, such as treatment in primary care settings or through conservative management approaches? Or did they forego care altogether,

potentially leading to unresolved health issues or adverse outcomes? Understanding these dynamics is essential to ascertain whether this reduction in care resulted in unmet medical needs or whether it represents an unintentional shift toward more appropriate care delivery. Such insights could provide valuable context for assessing whether these changes align with principles of necessity and proportionality in healthcare provision.

Moreover, the long-term effects of this vanished care must be studied in detail. Questions remain as to whether the reduced volume of both surgical and non-surgical procedures reflects changes on the demand side during crisis, for example, in disease incidence, patient preferences or systemic barriers to access. Alternatively, the crisis might have affected the use of care through supply side changes. Was the decline driven by restricted healthcare capacity: resource limitations, workforce shortages or a broader recalibration of clinical priorities during the pandemic? Understanding these drivers will inform strategies to balance immediate care demands with the need to ensure equitable access in future crises.

A better understanding might also help to define the appropriateness of care in non-crisis settings. The COVID-19 pandemic created a unique opportunity to critically evaluate which procedures are essential and which may be of limited benefit. However, the observed reduction in care must not be universally interpreted as progress towards more appropriate care.

### CONCLUSION
The COVID-19 pandemic significantly disrupted both surgical and non-surgical procedures within benign gynaecology pathways. Reduced care uptake during COVID-19 waves was not recovered, but forgone. The

reduction in surgical procedures did not correspond with increased use of non-surgical alternatives. This raises important societal questions about the consequences of unused care and especially whether the reduction in surgical procedures is desirable or has left patients with a disproportionately reduced quality of life.

Future research should prioritise evaluating the impact of this disruption at both patient and societal level. Qualitative research is needed to understand the individual patient experiences and outcomes, while quantitative studies should assess broader implications for care pathways and healthcare costs. Our findings offer valuable insights that can guide healthcare policy and practice, emphasising the importance of developing resilient systems capable of delivering high-quality care in the face of future disruptions and relative scarcity of medical capacity.

**Contributors** All authors made substantial contributions to the conception, design and acquisition of data for the work, as well as the analysis and interpretation of the data. ELMV and II had a particularly significant role in the analysis of the data. ELMV drafted the manuscript, with critical revisions from II, XK, RAdL and WJKH. All authors reviewed the manuscript critically for important intellectual content. Each author gave final approval of the version to be published and agrees to be accountable for all aspects of the work, ensuring that questions related to the accuracy or integrity of any part of the work are appropriately investigated and resolved. ELMV is the guarantor for this paper.

**Funding** This study was funded by ZonMw, the Dutch Organization for Health Research and Development, under project number 05160482240003.

**Competing interests** None declared.

**Patient and public involvement** Patients and/or the public were not involved in the design, conduct, reporting or dissemination plans of this research.

**Patient consent for publication** Not applicable.

**Ethics approval** This study involves human participants but was not approved by an Ethics Committee(s) or Institutional Board(s). We used CBS microdata for our research. As part of the data application process, CBS assesses whether our study complies with legal and ethical guidelines. This review ensures that our research meets ethical standards before access to the data is granted. Since our study is retrospective and based on fully anonymised data, it does not fall under the scope of the Medical Research Involving Human Subjects Act (WMO), and approval from an external Medical Ethics Review Committee (METC) is not required.

**Provenance and peer review** Not commissioned; externally peer reviewed.

**Data availability statement** Data may be obtained from a third party and are not publicly available. The data used for this study were obtained from Statistics Netherlands (Centraal Bureau voor de Statistiek, CBS) under a data-sharing agreement. Due to privacy regulations and the confidentiality of individual-level data, the dataset cannot be shared publicly. Access to CBS data is restricted and requires approval through their secure microdata services, available to registered institutions under specific conditions.

**ORCID iD**
Eva L M Velthuijs http://orcid.org/0000-0001-6386-4007

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
