## [Reviewer comments · BMJ Open]

ARTICLE DETAILS

Title (Provisional)

Substitution of Surgical Care within Benign Gynaecology during Covid-19: Waste of a Good Crisis? – A quantitative longitudinal study in the Netherlands

Authors

Velthuijs, Eva L. M.; Ismail, Ismail; Koolman, Xander; de Leeuw, Robert A.; Hehenkamp, Wouter J.K.

VERSION 1 - REVIEW

Reviewer	1
Name	Harris, John
Affiliation	University of Pittsburgh, Department of Obstetrics, Gynecology and Reproductive Sciences
Date	17-Mar-2025
COI	None

The authors present longitudinal trends in surgical and nonsurgical gynecologic care for 6 benign conditions (heavy menstrual blood loss, fibroids, endometriosis, prolapse, infertility, and first-trimester complications) between 2016 and 2022 in the Netherlands. COVID-19 was associated with an 18.3% drop in surgical and a 6% drop in non-surgical interventions, but nonsurgical procedures recovered by July 2020, but surgical procedures recovered much more slowly—only returning to baseline by 2022.

Changes in gynecology health services over time and how COVID19 affected the delivery of these services is of lasting interest to women's health specialists and general health services researchers, including readers of this journal.

This manuscript presents a large national dataset of gynecologic diagnoses and non-surgical and surgical interventions over a long time period. Unfortunately, the lack of more rigorous and extensive statistical hypothesis testing limits the impact and interpretation of this manuscript.

Comments

1. Abstract. Page 4, Line 40: Did age exclusions also applied to infertility and 1st pregnancy complication care as well?
2. Abstract. Page 5, Line 3-11: Results presented here lack levels of statistical significance (ie. P values or 95% CI). Results are not reported at any specific time period other than relative to 2020. It would be helpful to show trends at each of the four cohorts (prepandemic, initial pandemic, late pandemic, post pandemic. Reporting the results of the linear regression would be appreciated.
3. Introduction: I would recommend excluding the first paragraph of the introduction and start with known disruptions in gynecologic care as well as the specific policy interventions that happened (and when they were lifted) in the Netherlands. International readers (like me) are not familiar with the Dutch COVID19 policies.
4. Page 9, Line 50: Would you please describe whether DHD data contains 100% of Dutch healthcare claims, and if not, what populations are generally excluded or underrepresented?
5. Page 10, Line 28: Would you include in an appendix the definitions of each surgical and non-surgical intervention, including relevant DTC? What type of treatments were not considered surgical or non-surgical (Page 10, line 48-53)?
6. Page 12, Line 58: Please describe the model for the predicted trendline in more detail. What variables were included in the model?
7. Page 13, Line 19: Please include more data on this linear regression model including variables, clustering measures, amount of missing data, etc. Were other models than linear regression modeled (i.e., polynomials, etc.)? Were interactions tested? Typical difference-in-difference models used in longitudinal data include an interaction term between the natural experiment (pandemic) and the time variable.
8. Page 14: Table 2 is helpful, but not sufficient. Another table showing the yearly change in treatment types and the change between pandemic periods would be of help. Comparing both the baseline and the pandemic lows would be of help. Reporting p-values for these trends would be helpful throughout.
9. Are there other patient demographic, provider supply, or regional policy factors that partially explain these trends?
10. Discussion, Page 19: More attention should be paid to the relative rebound or lack thereof in care in 2021 and 2022. 2 sentences here barely delve into these important findings. For many readers, these 2021 and 2022 findings are much more interesting the 2020 findings, which were abnormal for many obvious reasons.
11. Figure 1-2: Would you show p-values between the predicted trends and actual findings? It would help to call the "trendline" a "predicated trendline" to show that it is a modeled outcome rather than a directly measured outcome

Reviewer	2
Name	Kelson, Zoe
Affiliation	University of Exeter, Mathematics
Date	06-May-2025
COI	None

This quantitative longitudinal study aims to examine the impact of the COVID-19 pandemic on the substitution of surgical procedures in benign gynaecology in the Netherlands.

Reviewer comments:

"The analysis identified a significant reduction in benign gynaecological care during 2020, with an 18.3% decrease in surgical procedures and a 6% decrease in non-surgical procedures across all pathways. Non-surgical procedures resumed pre-pandemic trends by mid-2020, whereas surgical procedures did not return to baseline levels." [Abstract]

Can statistical outcomes please be reported alongside study findings in the Abstract?

"Nationwide healthcare delivery was analysed across six benign gynaecological pathways from 2016 to 2022 using Vektis and Dutch Hospital Data (DHD), accessed via Statistics Netherlands (CBS)."

and

"Patients receiving care within these pathways between January 1, 2016, and December 31, 2022, were included"

Can the authors please confirm if this dataset can be considered to be representative?

"Pathway identification was based on Diagnosis Treatment Combinations (DTCs). Both surgical and non-surgical procedures were identified through healthcare products and activity numbers linked to individual patient records"

and

"To create our dataset, we had to combine several datasets from both DHD and Vektis within the Statistics Netherlands microenvironment. Combining, processing and cleaning of the datasets is done using RStudio."

Can the authors please comment on the completeness of this data?

"To account for seasonal variation and for graphical presentation only, a 3-month moving average was used to monthly absolute patient counts. Each point in the moving average represents the average of one preceding, the current and a subsequent month, to reduce noise"

Did the authors explore the effects of using different rolling averages?

"A predicted trendline for the years 2020 to 2022 was estimated based on the average annual growth from 2016 to 2019 combined with the 3-month moving average of 2019 patient counts."

Can the authors please specify what time series or other model was used to generate these extrapolations?

"Linear regressions were used to identify trends over time, with statistical significance set at $p < 0.05$. In the linear regression model, data from 2016 were used as the reference year. The year 2020 was included as a dummy variable to account for the impact of the COVID-19 pandemic. Additionally, a dummy variable for surgical care was included to differentiate between operative and non-operative care within the model"

Appropriate modelling methods have been applied.

Can the regression models and their outcomes please be shown in full, perhaps in the supplementary material?

Did the authors consider applying joinpoint regression analyses, using splines to model different time periods?

"No data imputation was performed for missing values"

Can the authors please confirm how much data was missing, and whether it can be considered to be missing at random?

"In 2020, the proportion of surgical care records declined across all pathways compared to 2019. The greatest decrease was in the prolapse pathway (-2.7%), while the smallest was in the infertility treatment pathway (-0.4%)."

Can these differences please be examined to determine if they are statistically significant?

"The drop in 2020 amounts to a reduction of 6% ($P=0.02$) for nonsurgical care group patient numbers and a reduction of 18.3% for surgical care ($P<0.01$ and compared to non-surgical care $P=0.01$)"

and

"In the care pathway heavy menstrual blood loss, the surgical procedures declined with 17.5% ($P<0.01$ and compared to non-surgical care $P=0.009$) while non-surgical procedures declined with 5.5% ($P=0.09$). In the uterine fibroids care pathway surgical care was reduced by 12.6% ($P<0.001$ and compared to non-surgical care $P=0.2$) and nonsurgical care was reduced by 7.5% ($P=0.01$). Both types of care in the endometriosis care pathway dropped in amount. Surgical care dropped with 16.9% ($P=0.1$ and compared to non-surgical care $P=0.7$) and non-surgical care dropped with 5.4% ($P=0.8$). Most decline is seen in the care pathway prolapse, with a decline of 10.4% ($P=0.7$) of non-surgical procedures and of 17.5% ($P=0.1$ and compared to non-surgical care $P=0.8$) of surgical procedures"

and

"The first trimester pregnancy complications care pathway shows a decline of only 0.6% (P=1) of non-surgical procedures over 2020 compared to predicted trends, while the surgical procedures declined with 17.7% (P=0.1 and compared to non-surgical care P=0.6). Infertility treatment shows a similar pattern with a decline of 2.1% (P=0.6) of non-surgical procedures and of 19.3% (P<0.001 and compared to nonsurgical care P=0.005) of surgical procedures"

Can it please be clarified in the Results text how these p-values were generated (i.e. what statistical technique was applied here)? Are these the outcomes of the Linear Regression analysis?

Can the authors please ensure statistically non-significant findings are not inferred as 'decline's?

"The observational nature of the study limits our ability to infer causality between the pandemic and the changes in care utilization. " [strengths and limitations of this study] and

"Nevertheless, there are important limitations. The observational nature of the study limits our ability to infer causality between the pandemic and the changes in care utilization. Additionally, we did not assess changes in primary health care use, which might have substituted a share of the secondary care, and patient outcomes, which restricts our ability to evaluate the effectiveness of the reduced or substituted non-surgical procedures. This latter limitation is particularly relevant when considering the long-term consequences of deferred surgical interventions"

A discussion on the study limitations is provided by the authors.

Thanks for providing a copy of the STROBE checklist.

VERSION 1 - AUTHOR RESPONSE

Reviewer: 1

Dr. John Harris, University of Pittsburgh

Comments to the Author:

The authors present longitudinal trends in surgical and nonsurgical gynecologic care for 6 benign conditions (heavy menstrual blood loss, fibroids, endometriosis, prolapse, infertility, and first-trimester complications) between 2016 and 2022 in the Netherlands. COVID-19 was associated with an 18.3% drop in surgical and a 6% drop in non-surgical interventions, but nonsurgical procedures recovered by July 2020, but surgical procedures recovered much more slowly—only returning to baseline by 2022.

Changes in gynecology health services over time and how COVID19 affected the delivery of these services is of lasting interest to women's health specialists and general health services researchers, including readers of this journal.

This manuscript presents a large national dataset of gynecologic diagnoses and non-surgical and surgical interventions over a long time period. Unfortunately, the lack of more rigorous and extensive statistical hypothesis testing limits the impact and interpretation of this manuscript.

Comments

1. Abstract. Page 4, Line 40: Did age exclusions also applied to infertility and 1st pregnancy complication care as well?

We thank the reviewer for this helpful clarification request. Age-based exclusions were applied consistently across all six care pathways for patients under the age of 18. In addition, a specific upper age limit (51 years) was applied only within the menstrual disorder care pathway. This criterion was introduced to exclude cases of postmenopausal bleeding, for which surgical treatment is typically not indicated and alternative treatment pathways do not apply. To improve clarity, we have revised the sentence in the abstract to read:

“Exclusions applied to all patients under 18 years old and, only within the menstrual disorder pathway, patients over 51 years old to exclude most postmenopausal blood loss cases where no alternative treatment applies.” (File “Manuscript”, section: ‘Abstract: participants’, page 3)

2. Abstract. Page 5, Line 3-11: Results presented here lack levels of statistical significance (ie. P values or 95% CI). Results are not reported at any specific time period other than relative to 2020. It would be helpful to show trends at each of the four cohorts (prepandemic, initial pandemic, late pandemic, post pandemic. Reporting the results of the linear regression would be appreciated.

We appreciate the reviewer’s valuable suggestion to strengthen the reporting of results in the abstract. To address this, we have revised the ‘Results’ section of the abstract to include statistical significance derived from the regression models. The revised sentence now reads:

“The analysis identified a significant reduction in benign gynaecological care during 2020, with an 18.3% ($P<0.01$) decrease in surgical procedures and a 6% ($P=0.02$) decrease in non-surgical procedures across all pathways. These differences were conducted using regression models that adjusted for time trends and the COVID-related impact.” (File: “Manuscript”, section: ‘Abstract: Results’, page 4)

3. Introduction: I would recommend excluding the first paragraph of the introduction and start with known disruptions in gynecologic care as well as the specific policy interventions that happened (and when they were lifted) in the Netherlands. International readers (like me) are not familiar with the Dutch COVID19 policies.

Thank you for this helpful comment. We agree that providing context on Dutch COVID-19 policies and their impact on surgical capacity is essential for international readers. In response, we have expanded the second paragraph of the introduction to include more detail on the timing and nature of pandemic-related policy interventions in the Netherlands, including triage protocols, periods of restriction, and the phased lifting of measures. The expanded paragraph now reads:

“ The COVID-19 pandemic, with its widespread disruption of elective surgical procedures and resource allocation toward urgent and critical care, created a natural experiment to examine whether substitution increased in times of acute and prolonged scarcity.[5-8] The initial COVID-19 period showed a reduction of 37% of all healthcare services across 20 countries.[9] Non-COVID hospital admissions were reduced by 28%. In the Netherlands, mainly non-cancer surgical treatments were cancelled during COVID-19 peaks.[10] National policies restricted non-urgent hospital care, especially during the first wave (March–May 2020), the second wave (October 2020–February 2021), and subsequent surges in 2021. Formal triage measures for surgical care, led by the Dutch Federation of Medical Specialists, were implemented to determine which procedures should be postponed.[11] Restrictions were lifted gradually during 2022 as hospital pressures subsided. The combination of a sharp reduction in surgical capacity and a lacking perspective to recover delayed care created a window of opportunity for less capacity consuming, less invasive (non-)surgical alternatives. While most research to date study the impact of the reduction of surgical care, none describe the effect of the pandemic on substitution.[9, 12-15]” (File “Manuscript”, section: ‘Introduction’, page 6)

To complement this, we have retained the first paragraph to frame the study within the broader context of the need for more sustainable and appropriate care, serving a wider context than the pandemic. We believe that the COVID-19 crisis functioned as a natural experiment that enabled observers of health care systems to study care efficiency and substitution and to reflect on how external shocks can drive health system transformation.

We hope that this revised introduction now provides both sufficient international context and retains the relevance of broader health system challenges.

4. Page 9, Line 50: Would you please describe whether DHD data contains 100% of Dutch healthcare claims, and if not, what populations are generally excluded or underrepresented?

Thank you for this valuable comment. We have clarified the scope and representativeness of the Dutch Hospital Data (DHD) in the revised manuscript. The DHD database captures nearly 100% of hospital-based care in the Netherlands, including admissions, diagnostics, and procedures, and is routinely used for national-level healthcare analyses. Combined with Vektis claims data—which includes all health insurance claims submitted to Dutch insurers—this dataset provides near-complete national coverage of hospital-based gynaecological care. While minor omissions may occur due to delayed processing or rare administrative inconsistencies, no systematic exclusions of population subgroups are known. We added the following sentences to clarify this point:

“These datasets together provide near-complete coverage of hospital-based gynaecological care in the Netherlands. DHD captures virtually all hospital claims data, while Vektis includes claims from all Dutch health insurers.” (File “Manuscript”, section ‘Methods: Data collection’, page 8&9)

5. Page 10, Line 28: Would you include in an appendix the definitions of each surgical and non-surgical intervention, including relevant DTC? What type of treatments were not considered surgical or non-surgical (Page 10, line 48-53)?

Thank you for your suggestion. We agree that the classification of interventions and the corresponding DTCs should be made transparent. We have clarified this in the revised methods section and now explicitly refer to Table 1, which provides an overview of the definitions and classification of surgical and non-surgical procedures, including the relevant DTC codes. We added

the sentence:

“Table 1 provides an overview of the classification and distribution of procedures within the selected DTCs.” to the method section and Table 1 follows after. (File: “Manuscript”, section: ‘Methods: data collection’ and Table 1. Distribution of procedures in surgical- or non-surgical care, page 9&10)

6. Page 12, Line 58: Please describe the model for the predicted trendline in more detail. What variables were included in the model?

Thank you for this important point. We have clarified the description of the predicted trendline model in the revised methods section. The predicted trendline was constructed as a univariate extrapolation based on historical patient volumes. Specifically, we calculated the average annual growth rate in monthly patient counts from 2016 to 2019 and applied this growth to a 3-month moving average of 2019 values to account for seasonality. No additional variables were included in this model, as the aim was to estimate a counterfactual trend based on pre-COVID historical growth alone. The revised sentences now read:

“A predicted trendline for the years 2020 to 2022 was estimated to serve as a counterfactual, representing a scenario without COVID-19. This prediction was constructed by calculating the average annual growth rate in patient volumes between 2016 and 2019, and applying this rate to the baseline monthly volumes of 2019. This approach is an extrapolation model based solely on monthly data in patient volumes.” (File: Manuscript, section: ‘Methods: Outcome measures’, page 11)

We hope this clarifies the construction of the counterfactual model.

7. Page 13, Line 19: Please include more data on this linear regression model including variables, clustering measures, amount of missing data, etc. Were other models than linear regression modeled (i.e., polynomials, etc.)? Were interactions tested? Typical difference-in-difference models used in longitudinal data include an interaction term between the natural experiment (pandemic) and the time variable.

Thank you for this thoughtful comment. We agree that regression models are often used to estimate adjusted associations or model functional forms. However, in our case, we used linear regression models primarily as a tool to quantify unadjusted differences between groups and to assess whether these differences reached statistical significance. Our intention was not to estimate causal or conditional effects, nor to model non-linear relationships or interactions. For this reason, we did not include interaction terms, polynomial terms, or adjust for covariates, as this would have suggested a level of causal inference we did not aim to make.

To avoid confusion about the role of these models in our analysis, we have revised the manuscript to clarify that our regression analyses were used to test raw associations rather than to control for confounding or to fit complex model structures. A table with results has now been added to the Results section. (File “Manuscript”, section: ‘Methods: statistical analysis’, page 12 and ‘Results: Table 3. Regression model output for yearly trends and COVID-related changes in surgical and non-surgical care across the six gynecological care pathways.’, page 16)

8. Page 14: Table 2 is helpful, but not sufficient. Another table showing the yearly change in treatment types and the change between pandemic periods would be of help. Comparing both the

baseline and the pandemic lows would be of help. Reporting p-values for these trends would be helpful throughout.

We thank the reviewer for this valuable suggestion. In response, we have updated Table 2 to include the year-on-year percentage change in treatment types for each pathway, along with the associated p-values. This allows for a clearer comparison between baseline years (2016–2019), the pandemic onset (2020), and the subsequent years (2021–2022). The revised table now offers a more granular overview of care trends over time and improves interpretability of changes both within and across the pandemic periods. We believe this enhances the clarity and robustness of our findings. (File “Manuscript”, section: ‘Results: Table 2. Baseline characteristics care pathways per year’, page 13)

9. Are there other patient demographic, provider supply, or regional policy factors that partially explain these trends?

We thank the reviewer for this thoughtful observation. We agree that demographic characteristics, provider supply, and regional or organizational factors are likely to have played an important role in shaping the observed trends in gynaecological care volumes. While our quantitative analysis is based on national-level registry data and does not include granular variables at the patient, provider, or regional level, we recognize the importance of offering plausible interpretations of the observed patterns.

In response to the reviewer’s suggestion, we have substantially revised several paragraphs in the Discussion section to improve clarity and enhance the presentation of explanatory hypotheses. We now more explicitly distinguish between patient-related factors (such as age-related hesitancy to seek care, symptom fluctuation, or natural resolution), organizational constraints (such as staff redeployment, reduced operating room capacity, and prioritization of urgent care), and the role of both clinician and patient preferences (e.g. perceptions of non-surgical alternatives, shared decisions to delay surgery). We also reflect on how these factors may have interacted to produce persistent changes, not only during the acute pandemic period but also in the years that followed.

Although our dataset does not allow for formal testing of these explanatory factors, we believe that a structured and transparent discussion of these mechanisms strengthens the interpretation of our findings and highlights important directions for future research. We have adjusted the relevant sections accordingly. (File: “Manuscript”, section: ‘Discussion’, pages 18, 19&20)

10. Discussion, Page 19: More attention should be paid to the relative rebound or lack thereof in care in 2021 and 2022. 2 sentences here barely delve into these important findings. For many readers, these 2021 and 2022 findings are much more interesting the 2020 findings, which were abnormal for many obvious reasons.

We appreciate this insightful comment and agree that the patterns observed in 2021 and 2022 merit more extensive discussion. We have therefore expanded this section to provide a deeper interpretation of the lack of rebound in care utilization. The revised paragraph now elaborates on potential explanations—including deferred or forgone care, mortality, symptom resolution (e.g., through menopause), and structural or organizational changes—while also highlighting the implications for long-term gynaecological care planning. We hope this offers a more nuanced interpretation of the post-pandemic period, in line with your helpful suggestion. (File “Manuscript”, section: ‘Discussion’, page 18&19)

11. Figure 1-2: Would you show p-values between the predicted trends and actual findings? It would help to call the “trendline” a “predicated trendline” to show that it is a modelled outcome rather than a directly measured outcome

We thank the reviewer for this helpful suggestion. In response, we have now added p-values to Figures 1 and 2 to indicate the statistical significance of the differences between the observed values and the predicted trendline for the year 2020. Additionally, we have revised the figure legends and labels to refer to the lines as “predicted trendlines,” clarifying that they represent modelled outcomes rather than observed data. Furthermore, the figure legends explicitly state that the reported p-values are based on the regression models presented in Table 3. (Files “Figure 1 V2 12-6-2025” and “Figure 2 V2 12-6-2025”)

Reviewer: 2

Prof. Zoe Kelson, University of Exeter

Comments to the Author:

This quantitative longitudinal study aims to examine the impact of the COVID-19 pandemic on the substitution of surgical procedures in benign gynaecology in the Netherlands.

Reviewer comments:

"The analysis identified a significant reduction in benign gynaecological care during 2020, with an 18.3% decrease in surgical procedures and a 6% decrease in non-surgical procedures across all pathways. Non-surgical procedures resumed pre-pandemic trends by mid-2020, whereas surgical procedures did not return to baseline levels." [Abstract]

Can statistical outcomes please be reported alongside study findings in the Abstract?

Thank you for your critical observation. We added statistical outcomes to the study findings in the abstract. Following your suggestion we changed the sentence to:

“ The analysis identified a significant reduction in benign gynaecological care during 2020, with an 18.3% (P<0.01) decrease in surgical procedures and a 6% (P=0.02) decrease in non-surgical procedures across all pathways.” (File: “Manuscript”, section: ‘Abstract: Results’, page 4)

"Nationwide healthcare delivery was analysed across six benign gynaecological pathways from 2016 to 2022 using Vektis and Dutch Hospital Data (DHD), accessed via Statistics Netherlands (CBS)."

and

"Patients receiving care within these pathways between January 1, 2016, and December 31, 2022, were included"

Can the authors please confirm if this dataset can be considered to be representative?

We thank the reviewer for highlighting the need to clarify the representativeness of our dataset. We confirm that the data used in this study are representative of hospital-based gynaecological care in the Netherlands. The Dutch Hospital Data (DHD) registry captures virtually all hospital care claims nationwide, while Vektis compiles claims submitted to all Dutch health insurers. Because our study focuses specifically on secondary care provided within hospitals, and both DHD and Vektis collectively cover nearly the full scope of this domain, we consider the dataset to be highly representative of the population and care pathways under investigation. We added the following sentences to clarify this point:

“These datasets together provide near-complete coverage of hospital-based gynaecological care in the Netherlands. DHD captures virtually all hospital claims data, while Vektis includes claims from all Dutch health insurers.” (File “Manuscript”, section ‘Methods: Data collection’, page 8&9)

"Pathway identification was based on Diagnosis Treatment Combinations (DTCs). Both surgical and

non-surgical procedures were identified through healthcare products and activity numbers linked to individual patient records"

and

"To create our dataset, we had to combine several datasets from both DHD and Vektis within the Statistics Netherlands microenvironment. Combining, processing and cleaning of the datasets is done using RStudio."

Can the authors please comment on the completeness of this data?

We confirm that the data extracted from DHD and Vektis are complete for the selected pathways and years within the CBS secure environment. For hospital-based care, data completeness is ensured through mandatory national reporting for claims and hospital activities. Data cleaning and linkage across datasets were performed using unique anonymized patient identifiers. No imputation was required. To clarify this, we included the following sentence in the manuscript:

"Due to mandatory national reporting, data completeness was high, and no imputation was needed." (File "Manuscript", section 'Methods: Data processing', page 11)

"To account for seasonal variation and for graphical presentation only, a 3-month moving average was used to monthly absolute patient counts. Each point in the moving average represents the average of one preceding, the current and a subsequent month, to reduce noise"

Did the authors explore the effects of using different rolling averages?

We thank the reviewer for this thoughtful suggestion. For graphical presentation, we primarily applied a 3-month moving average to the monthly absolute patient counts to reduce random month-to-month variation and improve interpretability. We also explored alternative window sizes, including a 2-month moving average and quarterly averages. The 2-month average appeared too volatile with more pronounced short-term fluctuations, while the quarterly average overly smoothed the data, obscuring important temporal changes—particularly around the COVID-19 period. Therefore, we concluded that the 3-month moving average provided the best balance between reducing noise and preserving meaningful trend changes. Importantly, all statistical analyses were performed on the original, unsmoothed monthly data.

"A predicted trendline for the years 2020 to 2022 was estimated based on the average annual growth from 2016 to 2019 combined with the 3-month moving average of 2019 patient counts."

Can the authors please specify what time series or other model was used to generate these extrapolations?

We thank the reviewer for this important point. In response, we have clarified the construction of the predicted trendline in the revised Methods section. The extrapolated trendline was developed as a simple univariate projection based on historical patterns in monthly patient volumes. Specifically, we calculated the average annual growth rate in patient counts between 2016 and 2019. This growth rate was then applied to the 3-month moving average of monthly volumes in 2019, which served as the baseline for extrapolation. No additional covariates were included, as our intention was to estimate a counterfactual trend assuming continuation of pre-pandemic growth, without the influence of COVID-19.

The revised manuscript now states:

"A predicted trendline for the years 2020 to 2022 was estimated to serve as a counterfactual, representing a scenario without COVID-19. This prediction was constructed by calculating the average annual growth rate in patient volumes between 2016 and 2019, and applying this rate to the

baseline monthly volumes of 2019. This approach is an extrapolation model based solely on monthly data in patient volumes.” (File: Manuscript, section: ‘Methods: Outcome measures’, page 11)

We hope this explanation sufficiently clarifies the modelling approach used to construct the counterfactual trendline.

"Linear regressions were used to identify trends over time, with statistical significance set at $p < 0.05$. In the linear regression model, data from 2016 were used as the reference year. The year 2020 was included as a dummy variable to account for the impact of the COVID-19 pandemic. Additionally, a dummy variable for surgical care was included to differentiate between operative and non-operative care within the model"

Appropriate modelling methods have been applied.

Can the regression models and their outcomes please be shown in full, perhaps in the supplementary material?

We thank the reviewer for this helpful suggestion. In response, we have added the complete output of the regression models to the main manuscript (Table 3) to enhance transparency and provide a full overview of the model specifications and results. This includes all relevant coefficients, interaction terms, and significance levels. To guide the reader, we have introduced the table with the following paragraph in the Results section:

“The percentages and p-values reported above are derived from regression models that adjust for time trends, intervention type (surgical vs. non-surgical), and the specific impact of the year 2020 (COVID-19). Full model coefficients and interaction terms are presented in Table 3.” (File “Manuscript”, section ‘Results: Table 3. Regression model output for yearly trends and COVID-related changes in surgical and non-surgical care across the six gynecological care pathways.’, page 16)

We hope this addition offers the requested clarity and completeness regarding the underlying statistical models.

Did the authors consider applying joinpoint regression analyses, using splines to model different time periods?

We appreciate the reviewer’s suggestion regarding joinpoint regression and spline modelling. However, consistent with our overall analytic approach, we did not aim to model structural breaks or changes in slope between specific time periods. Instead, we used simple linear regression models to quantify unadjusted, overall trends in care volume over time and to assess whether deviations from pre-pandemic trends were statistically significant. These models were not intended to capture non-linearities or to formally test segmented changes, but rather to descriptively assess raw differences in observed data.

As with the regression models discussed above, our goal was not to estimate adjusted or causal effects, nor to impose complex functional forms. We have clarified this purpose in the revised manuscript to ensure the descriptive nature of our analyses is clearly understood.

“ To assess raw associations we used linear regression for continuous outcomes. To express regression outcomes in percentages, all continuous outcomes were log-transformed prior to analysis. When the distribution of the error term was not normal, we used non parametric bootstrap to

compute p-values. Where relevant, year-to-year differences in proportions were tested using chi-square tests." (File: Manuscript, section: 'Methods: Statistical analysis, page 12)

"No data imputation was performed for missing values"

Can the authors please confirm how much data was missing, and whether it can be considered to be missing at random?

We thank the reviewer for this useful suggestion. We confirm that the extent of missing data in our dataset was minimal. As the data are derived from national administrative hospital and claims registries with standardized reporting procedures, missingness is rare and typically results from incidental administrative delays or processing issues. We did not perform imputation, as the low proportion of missing values was not expected to meaningfully affect the analysis.

To clarify this in the manuscript, we have added the following sentences to the Methods section:

"Due to mandatory national reporting, data completeness was high, and no imputation was needed."

(File "Manuscript", section 'Methods: Dataset processing', page 11), and

"Missing data were minimal and no imputation was performed." (File: Manuscript, section:

'Methods: Statistical analysis, page 12)

"In 2020, the proportion of surgical care records declined across all pathways compared to 2019. The greatest decrease was in the prolapse pathway (-2.7%), while the smallest was in the infertility treatment pathway (-0.4%)."

Can these differences please be examined to determine if they are statistically significant?

Thank you for this valuable comment. We have examined the statistical significance of these differences and have incorporated the corresponding results into Table 2. Additionally, we have revised the text in the Results section to reflect these significance findings:

"In 2020, the proportion of surgical care records declined across all pathways compared to 2019. The greatest decrease was in the prolapse pathway (-2.7%, $P < 0.001$), while the smallest was in the infertility treatment pathway (-0.4%, $P < 0.001$)." (File: "Manuscript", section: 'Results, Table 2. Baseline characteristics care pathways per year, and text', page 13&14)

"The drop in 2020 amounts to a reduction of 6% ($P = 0.02$) for nonsurgical care group patient numbers and a reduction of 18.3% for surgical care ($P < 0.01$ and compared to non-surgical care $P = 0.01$)"

and

"In the care pathway heavy menstrual blood loss, the surgical procedures declined with 17.5% ($P < 0.01$ and compared to non-surgical care $P = 0.009$) while non-surgical procedures declined with 5.5% ($P = 0.09$). In the uterine fibroids care pathway surgical care was reduced by 12.6% ($P < 0.001$ and compared to non-surgical care $P = 0.2$) and nonsurgical care was reduced by 7.5% ($P = 0.01$). Both types of care in the endometriosis care pathway dropped in amount. Surgical care dropped with 16.9% ($P = 0.1$ and compared to non-surgical care $P = 0.7$) and non-surgical care dropped with 5.4% ($P = 0.8$). Most decline is seen in the care pathway prolapse, with a decline of 10.4% ($P = 0.7$) of non-surgical procedures and of 17.5% ($P = 0.1$ and compared to non-surgical care $P = 0.8$) of surgical procedures"

and

"The first trimester pregnancy complications care pathway shows a decline of only 0.6% ($P = 1$) of non-surgical procedures over 2020 compared to predicted trends, while the surgical procedures declined with 17.7% ($P = 0.1$ and compared to non-surgical care $P = 0.6$). Infertility treatment shows a similar pattern with a decline of 2.1% ($P = 0.6$) of non-surgical procedures and of 19.3% ($P < 0.001$ and

compared to nonsurgical care P=0.005) of surgical procedures"

Can it please be clarified in the Results text how these p-values were generated (i.e. what statistical technique was applied here)? Are these the outcomes of the Linear Regression analysis?

Thank you for this important point. We confirm that the p-values presented in the results are derived from the regression models used in our analysis. To clarify this for the reader, we have now explicitly stated in the Results section that the p-values originate from these regression analyses. Furthermore, we have added the full model outputs as Table 3 in the Results section to increase transparency regarding the applied statistical methods. We added the paragraph:

"The percentages and P-values reported above are derived from regression models that adjust for time trends, intervention type (surgical vs. non-surgical), and the specific impact of the year 2020 (COVID-19). Full model coefficients and interaction terms are presented in Table 3." (File: "Manuscript", section: 'Results, Table 3. Regression model output for yearly trends and COVID-related changes in surgical and non-surgical care across the six gynecological care pathways', page 16)

Can the authors please ensure statistically non-significant findings are not inferred as 'decline's?

We thank you for this important comment. We have carefully revised the Results section to avoid interpreting statistically non-significant differences as definitive declines. Where appropriate, we now refer to such findings as "percentual decreases" or "non-significant reductions" to more accurately reflect the results. Statistically significant changes continue to be referred to as declines or increases. (File "Manuscript", section: 'results', pages 14&15)

"The observational nature of the study limits our ability to infer causality between the pandemic and the changes in care utilization. " [strengths and limitations of this study]
and

"Nevertheless, there are important limitations. The observational nature of the study limits our ability to infer causality between the pandemic and the changes in care utilization. Additionally, we did not assess changes in primary health care use, which might have substituted a share of the secondary care, and patient outcomes, which restricts our ability to evaluate the effectiveness of the reduced or substituted non-surgical procedures. This latter limitation is particularly relevant when considering the long-term consequences of deferred surgical interventions"

A discussion on the study limitations is provided by the authors.

Thanks for providing a copy of the STROBE checklist.

Reviewer: 1

If you have selected 'Yes' above, please provide details of any competing interests.: Not applicable

Reviewer: 2

If you have selected 'Yes' above, please provide details of any competing interests.: Not applicable

VERSION 2 - REVIEW

Reviewer	1
Name	Harris, John

Affiliation **University of Pittsburgh, Department of Obstetrics,
Gynecology and Reproductive Sciences**

Date **03-Jul-2025**

COI

Thank you for your responsive revisions to this manuscript. I have no other comments or revision suggestions.